# Differential effects of propofol and ketamine on critical brain dynamics

**Thomas F. Varley** [1,2¤]*, **Olaf Sporns**[1], **Aina Puce**[1], **John Beggs**[3]

**1** Psychological & Brain Sciences, Indiana University, Bloomington, Indiana, USA, **2** School of Informatics, Indiana University, Bloomington, Indiana, USA, **3** Department of Physics, Indiana University, Bloomington, Indiana, USA

¤ Current address: Psychological & Brain Sciences, Indiana University, Bloomington, Indiana, USA
* tvarley@iu.edu

**Data Availability Statement:** All of the data is available for free download from the NeuroTycho website: http://neurotycho.org/anesthesia-and-sleep-task.

## Abstract

Whether the brain operates at a critical "tipping" point is a long standing scientific question, with evidence from both cellular and systems-scale studies suggesting that the brain does sit in, or near, a critical regime. Neuroimaging studies of humans in altered states of consciousness have prompted the suggestion that maintenance of critical dynamics is necessary for the emergence of consciousness and complex cognition, and that reduced or disorganized consciousness may be associated with deviations from criticality. Unfortunately, many of the cellular-level studies reporting signs of criticality were performed in non-conscious systems (in vitro neuronal cultures) or unconscious animals (e.g. anaesthetized rats). Here we attempted to address this knowledge gap by exploring critical brain dynamics in invasive ECoG recordings from multiple sessions with a single macaque as the animal transitioned from consciousness to unconsciousness under different anaesthetics (ketamine and propofol). We use a previously-validated test of criticality: avalanche dynamics to assess the differences in brain dynamics between normal consciousness and both drug-states. Propofol and ketamine were selected due to their differential effects on consciousness (ketamine, but not propofol, is known to induce an unusual state known as "dissociative anaesthesia"). Our analyses indicate that propofol dramatically restricted the size and duration of avalanches, while ketamine allowed for more awake-like dynamics to persist. In addition, propofol, but not ketamine, triggered a large reduction in the complexity of brain dynamics. All states, however, showed some signs of persistent criticality when testing for exponent relations and universal shape-collapse. Further, maintenance of critical brain dynamics may be important for regulation and control of conscious awareness.

## Author summary

Here we explore how different anaesthetic drugs change the nature of brain dynamics, using neural activity recorded from sub-dural electrophysiological arrays implanted in a macaque brain. Previous research has suggested that loss of consciousness under anaesthesia is associated with a movement away from critical brain dynamics, towards a less flexible regime. When comparing ketamine and propofol, two anaesthetics with largely

**Funding:** TFV is supported by the NSF-NRT grant 1735095, Interdisciplinary Training in Complex Networks and Systems. The funders had no role in study design, data collection and analysis, decision to publish, or preparation of the manuscript.

**Competing interests:** The authors have declared that no competing interests exist.

different effects on consciousness, we find that propofol, but not ketamine, produces a dramatic reduction in the complexity of brain activity and restricts the range of scales where critical dynamics are plausible. These results suggest that maintenance of critical dynamics may be important for regulation and control of conscious awareness.

## Introduction

The hypothesis that the brain operates in a critical regime near a "tipping" point between different states (often, but not always, described as low and high entropy, respectively) is growing in popularity, both on neurophysiological evidence and due to appealing properties of critical, or near-critical, systems [1], that are thought to be key elements of optimal nervous system functioning. In both *in vivo* recordings and simulations, critical systems show the widest dynamic range [2–5], which indicates that critical systems can respond to, and amplify, a broad range of signals. For a system embedded in a complex environment, where salient signals may have a variety of intensities, a broad dynamic sensory range is a necessary adaptation. Critical systems also have optimized memory storage capacity from which complex information about states and patterns can be retrieved [6–9]. A related series of findings suggest that, in addition to optimal dynamic range and memory capacities, critical systems have optimized information transmission capabilities [10–12], and that the ability to integrate information is locally maximal near the critical zone as well [13]. Signs of critical dynamics have been found in the brains of a large number of different animals with different levels of CNS complexity, including humans [14], zebrafish [15], turtles [16], rat brain cultures [17], leeches [18], freely-behaving rodents [19], and non-human primates [20]. This suggests that the evolutionary advantage conferred by critical neural dynamics is highly conserved between species. The "critical brain" hypothesis has not been universally accepted however [21], and disagreement remains within the field as to when it is acceptable to conclude data was produced by a critical or near-critical system [17].

Despite the considerable work that has been performed on identifying indicators of criticality at the level of neuronal circuits, the relationship between critical dynamics (or lack thereof) at the micro-scale and macro-scale phenomena such as cognition, sensation, and awareness is less clear. In human neuroimaging studies, deviations from criticality have been found to be associated with altered or abnormal states of consciousness. Long-term sleep deprivation reduces signatures of critical dynamics in human MEG activity [14], and the pathophysiology of epilepsy has been modelled as a failure to maintain healthy critical dynamics [22, 23]. Psychiatric disorders such as schizophrenia, obsessive-compulsive disorder and major depressive disorders have also been discussed as possible deviaitons away from "healthy" critical dynamics. Human neuroimaging studies have suggested that "classical" serotonergic psychedelic drugs (eg. psilocybin, lysergic acid diethylamide, dimethyltryptamine, etc) increase markers of criticality, both in fMRI and MEG studies (for review see [24]). These findings have prompted some to hypothesize that normal consciousness emerges when brain activity is tuned near a critical regime, and that alterations in consciousness are reflective of transitions towards, or away from, the critical regime [25, 26].

However, signs of criticality in neural tissue are observed in animals (or cultured tissues) that could not be plausibly considered conscious. This includes dissociated cortical cultures [17], excised turtle brains [16], and animals anesthetized with a number of different anaesthetics [27]. Clearly criticality alone is not sufficient for conscious awareness, although the question of necessity remains open. Cortical recordings from anaesthetized rats found loss of

critical dynamics during the period of anaesthesia that re-emerged over the course of waking [28]. A similar study using cellular imaging found that anaesthesia reversibly altered signs of critical dynamics, as well as reducing the complexity of brain activity [29]. Computational models of brain dynamics informed by the pharmacology of anaesthetic agents have suggested that loss of consciousness induced by anaesthesia may be associated with a loss of critical dynamics [30] at a macro scale as well, although this remains an under-explored area of research.

To assess the relative differences in criticality between states of consciousness, we used publicly available invasive electrocorticography (ECoG) recordings from a *Macaca fuscata* monkey to compare the normal, awake resting-state (the monkey is awake, with eyes open, restrained in a primate chair) to two distinct states of anesthesia induced by two different drugs: propofol and ketamine (recordings were begun only after loss of consciousness had been diagnosed). These recordings are available as part of the NeuroTycho project's open data initiative [31, 32]. While both drugs induce surgical anaesthesia in high doses, and are used in clinical settings, they display markedly different pharmacologies and trigger different subjective experiences at low to moderate doses. Propofol is a commonly-used anaesthetic administered either by inhalation or intravenously. While its exact mechanism of action remains unknown, it is believed that it's primary action is through potentiating inhibitory GABA$_A$ receptors resulting in widespread decreases in neuronal activity [33]. Behaviourally, propofol induces sedation, atonia, and at high doses, cardiac arrest, respiratory depression, and hypotension [34]. Ketamine acts as an antagonist of glutamatergic NMDA receptors and, in contrast to propofol, causes mild nervous system stimulation and has little effect on respiration [34, 35]. Furthermore, while propofol induces a state reminiscent of deep coma, at sub-anaesthetic doses, ketamine induces an atypical state known as "dissociative anaesthesia" [36, 37], in which a person appears unresponsive to sensory or physical stimuli, but will often experience dream-like states, including hallucinations, out-of-body experiences, and immersive visions [35]. In this way, ketamine models other altered states of consciousness such as NREM sleep and locked-in syndrome, where phenomenological awareness can persist, despite an external appearance of unresponsive unconsciousness.

We chose electrophysiological recordings for this analysis for several reasons. Many previous studies that have found signs of criticality in anaesthetized animals have been performed while recording at the single-neuron micro-scale, while studies reporting alterations to critical dynamics following changes to consciousness have been performed using macro-scale neuroimaging methods (fMRI, MEG, etc). Intracranial ECoG provides a type of meso-scale signal: with coarse-graining of electrical activity in large numbers of neurons ($\approx 10^5$), but with finer localization than scalp EEG/MEG [38]. Consequently, these data are ideal for the study of criticality as the likely highly local preservation of critical dynamics in neuronal-level studies may interact with alterations to criticality seen in more macro-scale neuroimaging studies. Human ECoG studies have found indirect evidence of dynamical criticality [39, 40]—not using avalanche-based analyses such as we used here. Alonso et al (2014) reported changes high-frequency ECoG signal stability during the onset of clinical anaesthesia although signs of metastability persisted through all states.

Here we test two hypotheses concerning the relationship between consciousness and critical dynamics. Specifically, we test if:

1. during the awake state, activity would express several hallmarks of criticality, including a power-law distribution of avalanche sizes and durations, exponent relations [17], as well as signs of data collapse when subsampling channels (known as finite size scaling).

2. propofol would dramatically reduce signs of critical dynamics, but that criticality would persist under the influence of ketamine. This is based on the phenomena of "ketamine dreams" discussed above: based on the persistence of phenomenological consciousness under ketamine, we would signs of consciousness-like dynamics to persist under ketamine but not necessarilly under propofol. We also included a related measure, complexity [41], which has been hypothesized to relate to consciousness and found to be associated with criticality in cortical cultures [13].

For a glossary giving definitions for the various technical terms used in this paper, see Glossary.

## Materials and methods

### Ethics statement

All data used in this experiment was taken from an open-source repository (http://www. neurotycho.org/). The original data was collected "in accordance with the experimental protocols (No. H24-2-203(4)) approved by the RIKEN ethics committee and the recommendations of the Weatherall report, "The use of non- human primates in research" [32]. Readers are referred to the quoted paper, as well as [31] and the Neurotycho repository website.

### Data set and preprocessing

**Data set.** We used the NeuroTycho dataset, an open-source set of 128-channel invasive ECoG recordings in two Macaca fuscata monkeys [31]. For this study, we analysed the resting-state scans from one monkey (Chibi), as the scans from the other monkey contained intractable artifacts that could not be eliminated during preprocessing. Arrays were placed on the left hemisphere only, recording from all major areas including the medial wall (for a map, see Fig 1).

In both the propofol and ketamine conditions, the monkey was restrained in a primate chair and recorded during normal consciousness for 10 minutes, with a sampling frequency of 1 KHz. In the propofol condition, the monkey was given a single bolus of intra-venous propofol (5.2 mg/kg), until loss of consciousness was observed (defined as unresponsiveness to having their forepaws touched and/or unresponsiveness to having their nose tickled with a cotton swab). For the ketamine condition, a single bolus of intramuscular ketamine (5.1 mg/kg) was administered until loss of consciousness was observed using the above criteria. No maintenance anaesthesia was given following the initial induction. Following diagnosis of anaesthesia, 10-minute recordings were taken, sampled at 1000 Hz. We did not include any recordings from the transition period between wakefullness and anaesthesia. More detailed discussion of the drug administration protocols can be found in [32] and the Neurotycho data wiki (http:// wiki.neurotycho.org/Anesthesia_and_Sleep_Task_Details).

There were a total of four scans in the awake condition, and two each in the propofol and ketamine conditions (each anaesthesia condition having it's own associated awake scan).

**Preprocessing.** The data were initially examined visually in EEGLAB (v.14.1.2) [42] run in a MATLAB (v. 2018b, The Mathworks, Natick, MA) environment for major non-reoccurring artefacts. Some sample five-second plots of the selected ECoG channels can be seen in Fig 2. After inspection (no data were removed), we performed Independent Component Analysis (ICA) using the *runica* function in EEGLAB to extract minor artefacts [38, 42]. We ran the ICA with the full 128 possible components and, across all scans, removed an average of 4 components, corresponding to brief bursts of high-frequency noise. After the ICA and removal of components, the data were filtered in MNE Python (v. 0.19.2) with a low-pass filter at 120Hz,

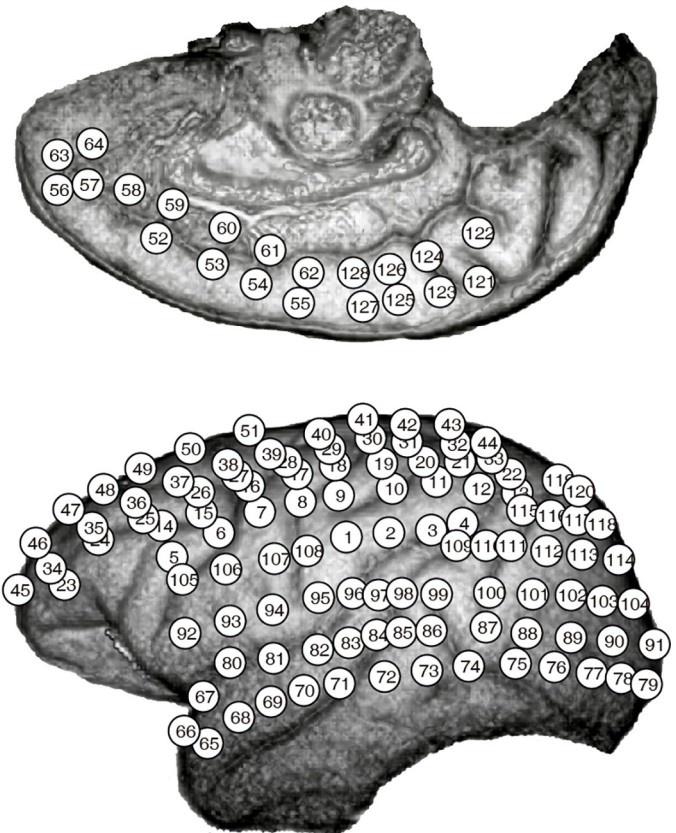

**Fig 1. Recording array map.** The placement of recording arrays for Chibi. Image taken from the NeuroTycho website, http://neurotycho.org/spatial-map-ecog-array-task.

high-pass filtered at 1Hz, and notch filtered at 50Hz and all subsequent harmonics up to 250 Hz, to account for electrical line-noise in Japan. We did not downsample the data, operating on the 1 KHz sample rate. All filters were of the FIR Overlap type (the default in MNE Python), [43])and all filters were run twice, forwards and backwards to eliminate phase-shift artefacts. Finally, we normalized the data by removing the mean of each channel and dividing it by its standard deviation. Due to the need for long time-series in this analysis, we operated on the

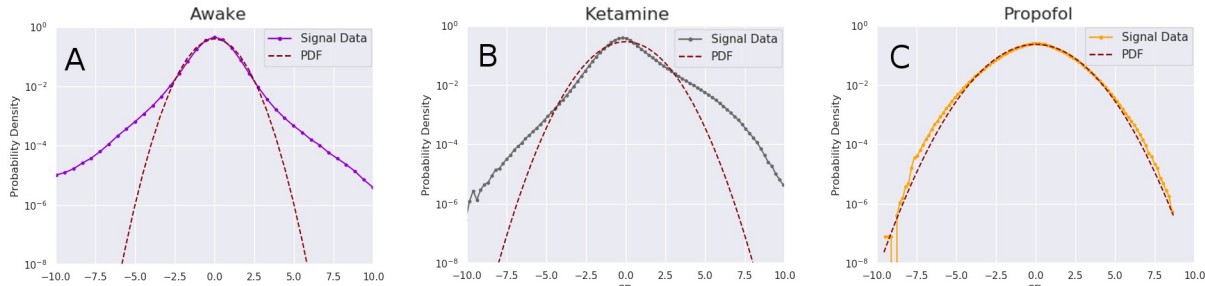

**Fig 2. Examples of raw EEG timeseries visualized.** Visualization of five seconds of all 128 channels for each condition. Top: awake, middle: ketamine, bottom: propofol. Images taken from EEGLAB (v.14.1.2) [42]. Note that the time-series are un-normalized, with scales ranging from 464-515 microvolts.

whole 10 minute time-series (there was no epoching). MNE Python (v. 0.19.2) was run in
Python 3.7.4 using the Anaconda (v. 3.7) environment.

## Point process

For each condition, we followed the method for point-processing data described by [44] and
[14]. Briefly: after filtering, the probability distribution of signal amplitudes in terms of their
standard deviation from the mean, was plotted, along with the best-fit Gaussian distribution,
and the point at which the actual distribution and the best-fit curve diverged were chosen as
the threshold ($\sigma$). This ensured that diversions above the threshold are unlikely to be due to
noise inherent in the signal. While there were differences in the point of divergence between
conditions, we chose a threshold of 4, as the most conservative likely value. We sampled a
range of thresholds around 4 (3 to 4.25) and found that this did not substantively alter the
results, although a too-low threshold ($< 3$) allowed excessive noise through, while a too-high
threshold ($> 4.5$) did not allow enough events for analysis of avalanches. For visualization of
the various distributions, see Fig 3.

Once the threshold had been chosen, all instances where the absolute value of a signal was
less than the threshold were set to zero:

$$\forall t \in |X| : X(t) = 0 \; if \; |X(t)| < \sigma$$

For all excursions above $\sigma$, the global maxima of the period was set to 1 and all other
moments set to 0. We calculated the Pearson correlation coefficient $\rho$ from the moment the
series crossed the $\sigma$ threshold ($t_{min}$) to the end point, the moment it dropped below $\sigma$ ($t_{max}$)
against the same range in every other channel, and if $\rho \geq 0.75$, we also set the local maximum
of the interval in the associated channel to 1 (even if it did not cross our threshold $\sigma$) as well
(sending all other values to 0). For a more in-depth discussion of this, see [14].

The resulting calculation produced a binary raster plot (channel × samples, for an example
see Fig 4) where every event corresponds to the moment of maximal excursion from the mean.
This data structure matches the type used when analysing spiking activity from cultured neu-
rons [10], making it amenable to many of the tools already developed for criticality analysis.
We chose not to re-bin the data, instead maintaining the 1000 Hz sampling rate across all
scans due to the relative shortness of the recordings: significant re-binning would reduce the
available data to the point that not enough individual avalanches could be identified.

## Avalanches and critical exponents

"Avalanches", defined as transient periods of coordinated activity between elements of a sys-
tem, are a feature common to many complex, dynamic systems [45]. In the context of the ner-
vous system, avalanches typically refer to synchronized action potentials in a neural network
[10]. Coordinated avalanches of activity have also been found in human EEG [44], MEG [14,
46], and fMRI [47] data. Avalanches are described by two values: the avalanche size ($S$, the
number of elements that participate in the avalanche) and the avalanche duration ($T$, the life-
time of the avalanche from start to finish). In critical systems, these two values are expected to
follow power laws with exponents $\tau$, and $\alpha$:

$$P(S) \propto S^{-\tau}$$
$$P(T) \propto T^{-\alpha}$$

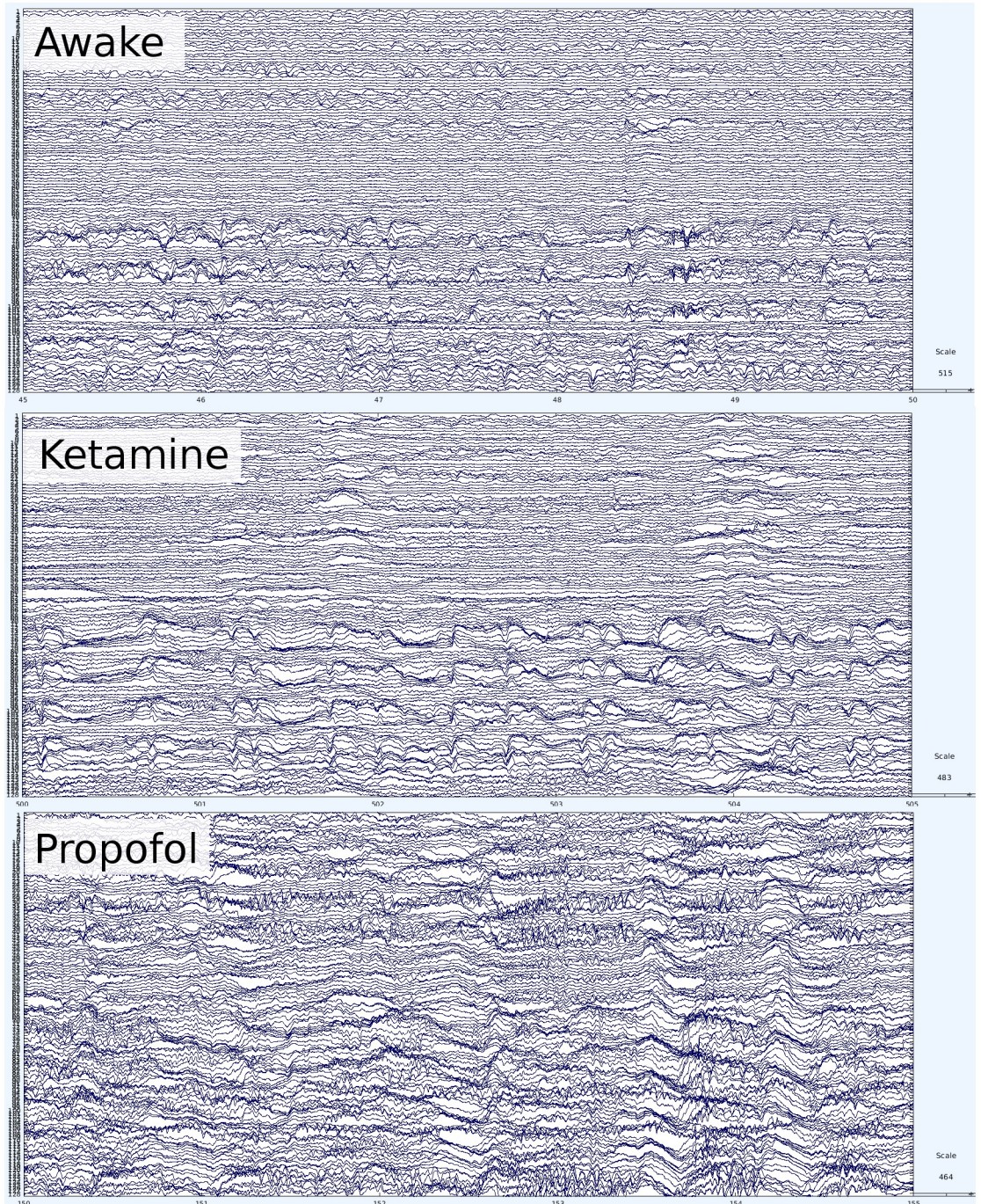

**Fig 3. Best-fit normal distributions between conditions.** Representative examples of the histograms of instantaneous amplitudes across the three conditions. **A**: the awake condition. **B**: ketamine, and **C**: propofol. Note that while ketamine allows for a heavy-tail to persist, the propofol condition collapses to a tight Gaussian distribution.

$S$ and $T$ should be distributed such that, over all avalanches, the average size for a given duration also follows a power law:

$$\langle S \rangle (T) \propto T^{1/\sigma v z}$$

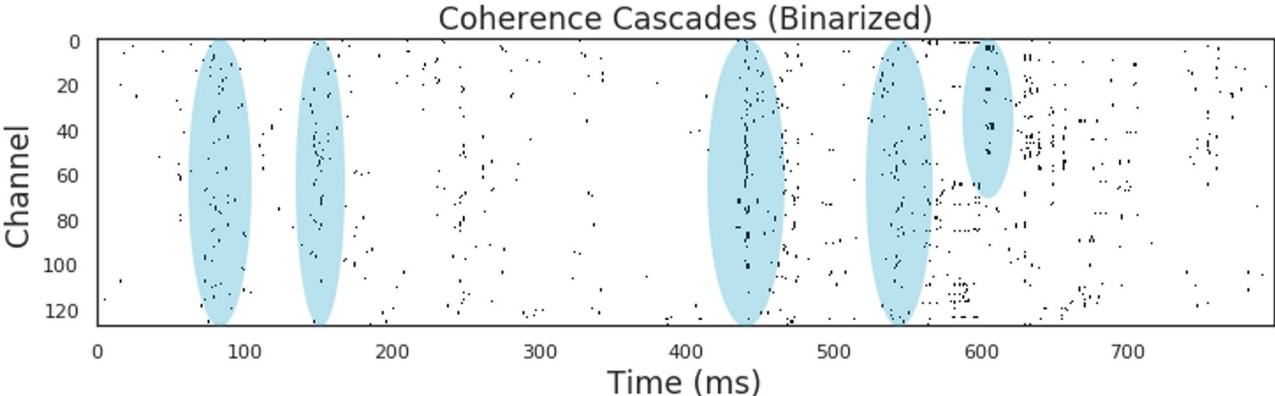

**Fig 4. Coherence cascades in the awake condition.** An example of a section of the raster plot. Notice the banding effect of coherence cascades (a few examples are highlighted by blue ovals). This banding effect shows the cascades of events that include many distinct channels that "fire" in synchrony.

$\tau$, $\alpha$ and $1/\sigma vz$ are collectively known as the critical exponents of the system [17]. We note that in this context, $1/\sigma vz$ is treated as a single variable and should not be considered a function of three distinct variables. Furthermore, all three exponents should be related to each other such that:

$$\frac{\alpha - 1}{\tau - 1} - \frac{1}{\sigma vz} = 0$$

It is generally considered to be insufficient evidence of criticality if only the distributions of avalanche sizes and durations follow power laws: these exponents must be related to each other through an exponent relation as given above [17].

To calculate the scaling exponents $\tau$ and $\alpha$, we used the NCC Toolbox (v. 1.0) [48] to extract avalanches from our binary time series, and perform a maximum likelihood estimate (MLE) of the power-law exponent [48, 49]. The NCC Toolbox returns several values associated with each power-law inference: the MLE value of the exponent, the minimum and maximum values of $x$ for which the power law estimate holds ($x_{min}$ and $x_{max}$, where $x$ can refer either to avalanche sizes or durations), and the $p$-value. It is crucial to note that $x_{min}$ and $x_{max}$ do *not* refer to the smallest and largest values of $x$ in the empirical distribution, but rather, the minimum and maximum values between which a power-law fit plausibly holds. Following the convention of Timme et al., (2016), we set our significance threshold such that we would only accept the power law hypothesis at $p \geq 0.2$ (this is the reverse of how significance estimation is usually performed, for discussion, see [49]. For the estimate of $1/\sigma vz$, we plotted $\langle S \rangle(T)$ against $T$ and extracted an estimate of the exponent by linear regression in log-log space. While this is a much cruder method than the MLE power law fit described above, unfortunately the values of $\langle S \rangle(T)$ vs. $T$ do not describe a probability distribution and so the usual methods of inference do not work.

When plotting the distribution of avalanche sizes and avalanche durations, we use complementary cumulative distribution functions (CCDFs) (defined as 1-CDF) instead of probability density functions (PDFs), following [50]. When dealing with power law (or plausibly power law) distributions with defined upper and lower bounds ($x_{min}$, $x_{max}$), the CDF and CCDFs may not display the characteristic straight line when plotted in log-log space [51, 52]. Consequently, distributions may look curved while still being plausibly drawn from a doubly-truncated power law. For a visualization of this, see Fig 5.

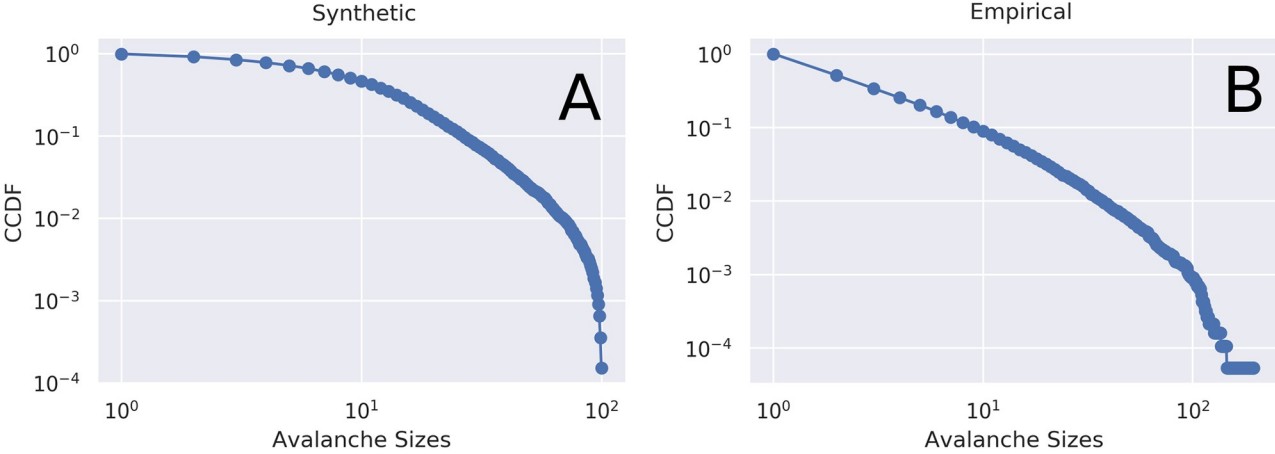

**Fig 5. Synthetic and empirical doubly-truncated power law distributions for the data. A**: an empirical, doubly-truncated power-law distribution taken from this dataset with an $x_{min}$ of 16 and an $x_{max}$ of 196. Within this range, the exponent has been estimated to be 2.7. **B**: a synthetic doubly-truncated power-law distribution with the same minimum and maximum values as the empirical distribution as well as the same exponent. Note that the synthetic model has significant curves which begin well before the upper-bound cut-off. When modelling a system that produces data within a fixed range, even power-law distributed values may not follow the canonical straight line when plotted in log-log space [51, 52].

**Avalanche shape collapse.** In addition to having sizes and durations, avalanches also have *profiles*, which describe the number of channels active at each moment over the course of the avalanche's lifetime. Near the critical regime, the profiles of all avalanches should be self-similar, that is, it should be possible to find some scaling exponent $\gamma$ such that all avalanches can be rescaled to lie on top of one another [13, 17]. This is referred to as "shape collapse" and is a commonly used indicator of operating near the critical point in dynamical systems.

This scaling parameter $\gamma$ can be defined as:

$$\gamma = \frac{1}{\sigma vz} - 1$$

where $1/\sigma vz$ is the same exponent that describes the distribution of average avalanche sizes for a given duration.

To calculate shape collapse, we first averaged together all avalanches of a given duration to create "average profiles". From there, we used the NCC Toolbox [48] to find the optimal rescaling value and extract an estimate of $1/\sigma vz$.

## Data collapse and finite size scaling

A common issue with analysis of critical dynamics in complex systems is one of sub-sampling [53, 54]. When a very large system is sub-sampled, the characteristic power-law behaviour can be lost as large events are fragmented and perceived as separate, smaller events. This is a particularly salient issue in electrophysiological recordings, as the number of channels in an array is orders of magnitude less than the number of functional units in the cortex that could be participating in critical avalanche dynamics.

In a system near the critical point, while subsampling destroys the power-law, it should not destroy the scale-free nature of the distributions—that is, when appropriately renormalized, the distributions (despite not following power laws), should display data collapse. This property is known as finite size scaling. The logic here is the same as when performing data collapse on individual avalanches, as above.

We tested for distribution shape-collapse following a similar approach to a previously procedure [54]. For each scan, we randomly sub-sampled half, a quarter, and an eighth of the channels, ten times each, and from each, calculated the associated probability distribution of avalanche events and sizes. In addition to plotting them to visualize the differences between the four distributions (the original plus the three subsampled distributions), we calculated the average pair-wise Wasserstein metric [55] to quantify the similarity between all distributions.

We then performed the same sub-sampling again, this time rebinning the binary timeseries to reflect the sampling of elements: when taking half the channels, we rebinned by a factor of two, when taking a quarter of the channels, we rebinned by a factor of four, etc. If the system is poised near the critical point, after rebinning, the distributions of avalanche activities should collapse onto one-another. When recalculating the Wasserstein metric, the average pairwise distance should be significantly reduced.

## Complexity

Here, "complexity" of an N-dimensional system X ($C_N(X)$) can be intuitively understood as "the degree to which the whole is greater than the sum of it's parts" [41]. To calculate the complexity, the entropy of the entire system is compared to the joint entropies of all possible subsets of the system. Each state $i$ of an N-dimensional system can be described as an N-dimensional vector $x_i$, and the probability of state $i$ ($p_i$) is the number of occurrences of $i$ divided by the number of samples. The Shannon entropy ($H(X)$) of the system is given by:

$$H(X) = -\sum_i p(x_i) log_2(p(x_i))$$

We can measure the level of integration (or coordination) of a set of elements of the system by comparing the joint entropies of all the elements in the set to the sum of their individual entropies:

$$I(X)_j^k = \left(\sum_{j' \in k} H(X_{j'}^1)\right) - H(X)_j^k$$

Here, $k$ is the number of neurons in a subset, and $j$ is the index of a given subset of $k$ neurons in the set of all sets of $k$ neurons. This has also been called "multinformation" or "total correlation" [56]. For instance, $X_j^1$ refers to the $j^{th}$ element from the set of N -choose-1 elements alone, while $X_j^4$ refers to the $j^{th}$ set of 4 elements from the set of N-choose-4 sets of 4 elements.

The complexity is then found by comparing the integration of the entire system ($X_1^N$) to the average integration ($\langle I(k)_1^j \rangle j$) over every possible subset, for every possible subset size of $k$.

$$C_N(X) = \frac{1}{N}\sum_{k=2}^{N}\left[\left(\frac{k-1}{N-1}\right)I(X) - \langle I(X_j^k)\rangle_j\right]$$

This value of $C_N(X)$ is a more intuitively meaningful representation of "complexity" than other commonly used measures, such as Lempel-Ziv Complexity [57], in that it is low for systems that are both perfectly ordered and systems that are perfectly random [13], while peaking in systems that combine elements of both. Calculating the complexity of a system is a nontrivial task, exploding to intractable levels as $N$ gets even modestly large (for context, for a 128-channel system, it would take longer than the expected lifetime of the universe of exhaustively search all partitions). To avoid interminable run-times, the NCC Toolbox [48] includes

several corrections for sub-sampling and heuristics for estimating integration in large systems. One correction is to only consider those bins where at least one "event" occurs, consequently calculating the complexity of the avalanches themselves, which controls for variable numbers and distances between avalanches. Furthermore, the NCC Toolbox corrects for sub-sampling biases in the joint probability distribution by comparing the empirical integrations to an ensemble of time-randomized null models and subtracting the expected value of the distribution of null integrations and optimizing on the subsets of size $k$ that are most informative about the structure of the system. By implementing these corrections, the toolbox is able to infer the multi-scale integration/segregation structure without having to brute-force all possible bipartitions of a sparse multi-dimensional time-series.

As with the avalanche size and duration distributions, we used the same 30-second jitter null-model to explore the effect of randomizing the data. We hypothesized that jittering the data should significantly reduce the nonlinearity and total integrated information.

## Results

### Channel activity

We found large differences between the channel rates following the administration of both ketamine and propofol. In the awake condition, the average channel-wise firing rate was $9 \times 10^{-4}$ spikes per millisecond (s/m), which dropped to $5 \times 10^{-4}$ (s/m) in the ketamine condition and $8 \times 10^{-5}$ (s/m) under propofol. We found that, within conditions, there was a high correlation between channel activity rates during different scans. Within the awake condition (across all six possible pairwise comparisons), we found an average correlation of $0.89 \pm 0.03$. In the ketamine condition (which only allows a single comparison), the correlation was 0.76 and in the propofol condition, the correlation was 0.62). Interestingly, we did not find strong correlations when comparing channel activity rates within the same scan before and after induction of anaesthesia. The correlation between channel activity rates before and after ketamine induction was 0.225 (p-value $> 0.05$) and the correlation between channel activity before and after propofol induction was 0.06 (p-value $> 0.05$). These results suggest a much higher degree of dynamical similarity within conditions scanned on different days than between conditions induced during a single scan. For visualization of these results, see Fig 6.

Using the NeuroTycho channel labelling (see Fig 1), we found that, in the awake condition, high levels of activity were seen occipito-temporal and parietal regions of the brain. This pattern was disrupted by both propofol and ketamine. In the ketamine condition, there was a decrease in the range of channel activities, and in both anaesthetic conditions, there was a shift in activity towards the occipital lobe. This significance of this is unclear, as the monkeys were wearing eyeshades throughout the anaesthetic experience and consequently were not receiving visual input.

### Avalanche distributions

There are clear differences between the awake, ketamine, and propofol distributions of avalanche sizes and durations (Fig 7). The Wasserstein distances between the conditions reveal that, in all cases, the awake condition was far more similar to the ketamine condition than the propofol condition. The distance between the avalanche size distribution in the awake condition and the ketamine condition was 0.46, while between the awake condition and the propofol condition it was 1.77. The same pattern was apparent between the avalanche duration distributions. The distance between the awake and ketamine conditions was 0.05, while for the awake and propofol conditions it was 0.43 (for the raw values see Table 1).

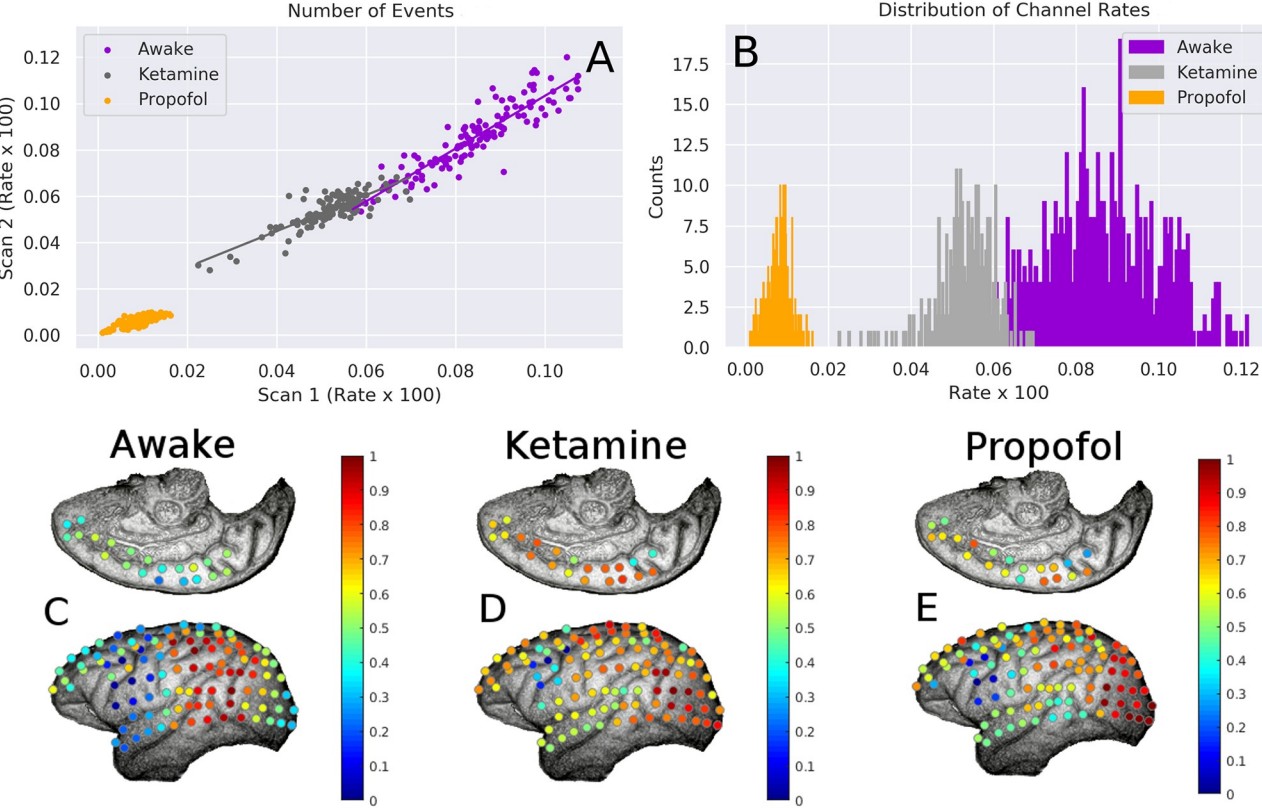

**Fig 6. Changes in channel activity levels between conditions. A**: For each condition, the correlation between channel activity (firing rate × 100) between two scans. In all conditions, it is apparent that those channels that are more activity in one scan are similarly active during different scans within the same condition. This suggests that brain dynamics are consistent within conditions. **B**: histograms of the the channel rate × 100 for all three conditions. It is apparent that propofol decreases the overall channel firing rate more that ketamine, although both have a lower firing rate than the awake condition. **C-E**: normalized channel activity rates projected onto the recording array. It is clear that propofol and ketamine disrupt the activity-rate structure that exists in the awake condition. Original images taken from the NeuroTycho wiki: http://neurotycho.org/spatial-map-ecog-array-task.

## Scaling exponents

For every distribution, there was a range between some $x_{min}$ and $x_{max}$ for which the power-law hypothesis held with $p \geq 0.2$. However, in the propofol condition, this range was dramatically restricted. In the awake condition, the average value $x_{min}$ for the avalanche size distribution was 16.25 channels and the average value for $x_{max}$ was 378.5 channels. In the ketamine condition, the average value $x_{min}$ for the avalanche size distribution was 14 channels, and the average value for $x_{max}$ was 347 channels. Finally, in the propofol condition, the average value $x_{min}$ for the avalanche size duration was 2 channels and the average value $x_{max}$ was 48 channels. The same pattern was observed in the distributions of avalanche durations. The average $x_{min}$ of the distribution of avalanche durations in the awake condition is 8 ms, while the average $x_{max}$ is 42.5 ms. In the ketamine condition, the average $x_{min}$ was 6 ms and the average $x_{max}$ is 42 ms. In the propofol condition, the range was much more constricted: the average $x_{min}$ was 2 ms and the average $x_{max}$ was 12 ms. These results indicate that, while all distributions had regions that could be plausibly modelled as following a power-law, in the propofol condition, this region was extremely narrow and typically restricted to very small values (excluding the tail, where the power-law distribution is most relevant). In contrast, ketamine maintained a range comparable to the awake condition, suggesting that, unlike propofol, ketamine anaesthesia does not suppress the propagation of large bursts of coordinated activity.

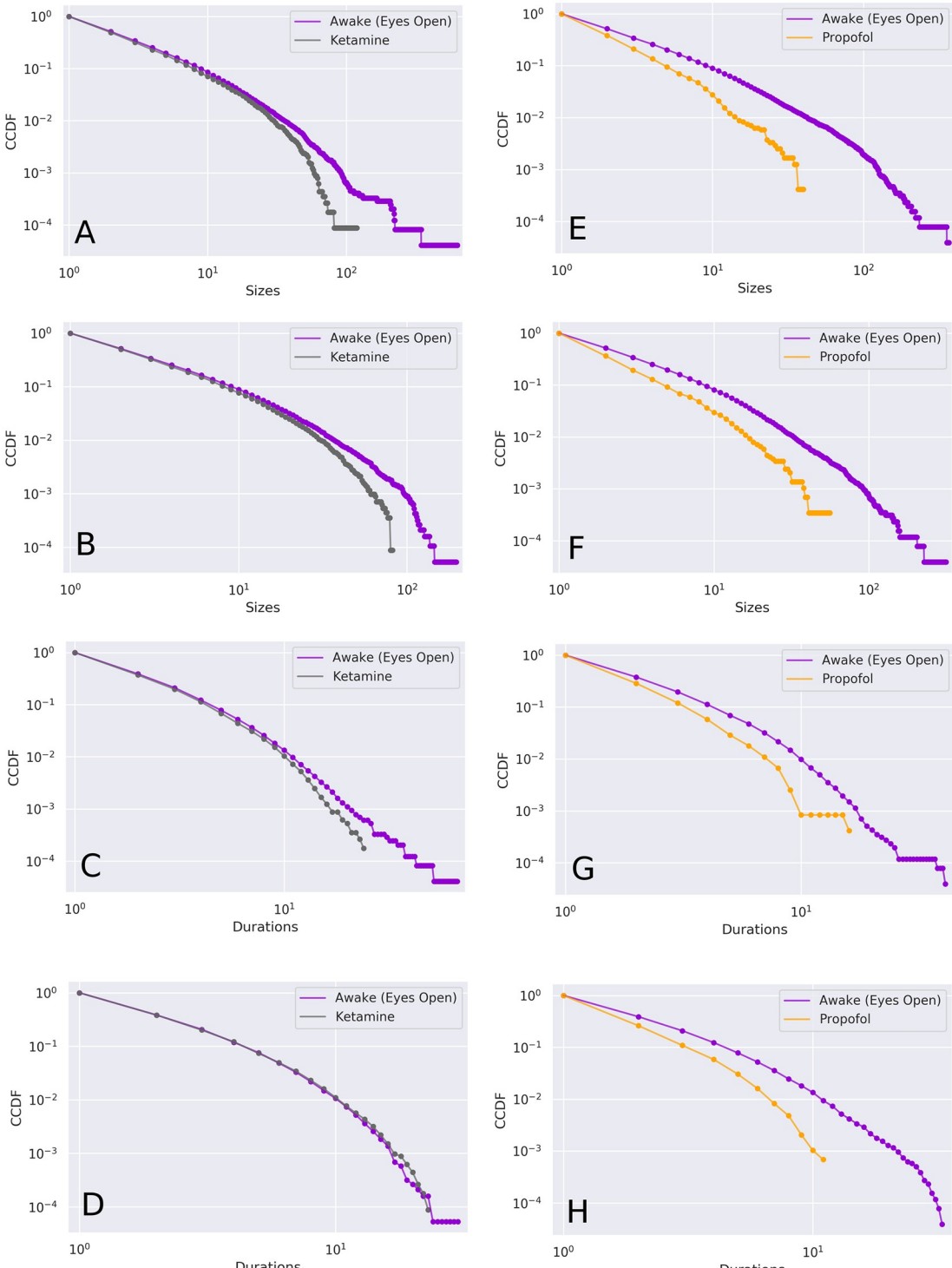

**Fig 7. CCDFs of Avalanche Size and Duration by Condition.** Comparison CCDFs for awake (purple) vs. ketamine (grey) and propofol (orange). Each plot includes two recordings: the anaesthesia condition (propofol or ketamine) and the associated pre-anaesthesia awake condition from the same monkey during the same session. **Right** panels: awake vs. ketamine. **Left** panels: awake vs. propofol. Avalanches sizes (top four panels) and durations (bottom four panels). Visual inspection shows that the ketamine condition tracks the awake condition much more closely than the propofol condition does: note how the propofol condition begins to drop below it's associated awake distribution almost immediately, while the ketamine distribution tracks the awake distribution for a considerable range. This indicates that ketamine supports larger, longer-lived avalanches than propofol.

**Table 1. Table of results.** A table giving all the raw values of each of the measures reported here, for each monkey, in each condition.

| Subject | Condition | $\tau$ | $x_{min}$ S | $x_{max}$ S | $\alpha$ | $x_{min}$ D | $x_{max}$ D | Exp. Rel. | Avg. Size / Dur | $1/\sigma\nu z$ | Complexity | Rebin KS D | No Rebin KS D | Rebin KS S | No Rebin KS S |
|---|---|---|---|---|---|---|---|---|---|---|---|---|---|---|---|
| 813KT | awake | 2.709 | 16 | 196 | 4.422 | 8 | 30 | 2.00234055 | 1.2529880263 | 1.144 | 0.0473942748 | 0.0444209518 | 0.1290998009 | 0.0623528147 | 0.1503449156 |
| 813KT | Ketam | 2.225 | 6 | 60 | 3.007 | 4 | 17 | 1.6383673469 | 1.2472912656 | 1.221 | 0.0316139101 | 0.0259019459 | 0.1375173692 | 0.0508161703 | 0.1653117455 |
| 802PF | awake | 2.484 | 13 | 368 | 4.783 | 9 | 41 | 2.5491913747 | 1.2654549163 | 1.262 | 0.0469241472 | 0.039505499
6 | 0.1244217247 | 0.0677982829 | 0.1588397862 |
| 802PF | Propo | 2.295 | 2 | 40 | 3.407 | 3 | 16 | 1.8586872587 | 1.2407987895 | 1.271 | 0.0036012452 | 0.029745315
5 | 0.1134300394 | 0.0788062334 | 0.1661862264 |
| 730PF | awake | 2.834 | 14 | 316 | 3.679 | 7 | 32 | 1.4607415485 | 1.3887947163 | 1.377 | 0.0470132706 | 0.0378105651 | 0.1299924942 | 0.0556711293 | 0.1639426327 |
| 730PF | Propo | 2.299 | 2 | 56 | 2.446 | 1 | 8 | 1.1131639723 | 1.4430822801 | 1.508 | 0.0051650643 | 0.0252161665 | 0.0926269857 | 0.0729665868 | 0.1340633845 |
| 719KT | awake | 2.93 | 22 | 634 | 4.029 | 8 | 67 | 1.5694300518 | 1.405535723 | -1.1 | 0.0477691165 | 0.0346876135 | 0.12823451 | 0.063741846 | 0.1690890631 |
| 719KT | Ketam | 2.93 | 22 | 634 | 4.029 | 8 | 67 | 1.5694300518 | 1.2537008081 | 1.25 | 0.029773869 | 0.0261199899 | 0.1324190428 | 0.0550677446 | 0.1445344283 |

For each range where the power-law fit held, we calculated the associated exponents, $\tau$ and $\alpha$. The sample size is too small for significance testing, although on average, the awake condition had the largest exponents for avalanche sizes (2.74) and durations (4.23), followed by ketamine, for both sizes (2.58) and durations (3.52) as well. Propofol had the smallest scaling exponents for both avalanche parameters (size: 2.3, durations: 2.93). The average calculated values of $1/\sigma vz$ were quite similar between all three (awake: 1.89, ketamine: 1.6, propofol: 1.5).

In all conditions, within the relevant ranges of $x_{min}$ and $x_{max}$, we found a strong relationship between the average size for a given duration (see Fig 8). The calculated value of $1/\sigma vz$ was similar across all conditions (awake: 1.33, ketamine: 1.25, propofol 1.34). Having two separate estimations of $1/\sigma vz$ (one from the exponent relation, one from the average size for a given duration) allows us to see how well the scaling exponents relate. Surprisingly, the awake condition had the greatest percentage difference between the two values, at 35.19%, followed by ketamine at 24.76% and finally propofol with 10.83%. We had hypothesized that the awake and/or ketamine conditions would show the highest degree of concurrence between the measures, but instead, the awake condition has the lowest degree of concurrence. The significance of this is difficult to explain. One possibility is that the awake condition is noisier than either anaesthesia condition, which would reduce the critical fitness, as well as driving down the $x_{min}$ value below which a power-law holds (see Discussion for more on this issue). Finally, we note that, while concurrence is highest in the propofol condition, it is over a much narrower range of $x$ values for both avalanche sizes and durations, as opposed to the awake and ketamine conditions and so the higher concurrence may be reflective of the more restricted range with fewer degrees of freedom (for further discussion of this issue, see the Discussion).

## Avalanche shape collapse

The avalanche collapse results are largely consistent with the analysis of average size for a given duration. The estimated values of $1/\sigma vz$ calculated from the shape collapse between conditions were largely similar: awake: 1.261, ketamine: 1.24, propofol 1.39. Once again, the percentage difference between these values and the exponent relationship was highest in the awake condition (40.2%), followed by the ketamine condition (25.95%), and with the greatest agreement in the propofol condition (6.7%). We also calculated the percentage difference between values of $1/\sigma vz$ derived from the shape collapse and the average size/given duration analysis. These showed a much higher degree of concurrence. The awake condition again had the greatest percentage difference (5.19%), followed by propofol (3.48%), and ketamine had the lowest percentage difference (1.2%).

## Finite-size scaling and distribution collapse

In addition to power-law and exponent-relation based indicators of critical dynamics, we also tested whether the probability distributions of avalanche sizes and durations exhibited data collapse when subsampled (finite size scaling) [54]. We initially sub-sampled the system by randomly selecting subsets of channels (half the channels, a quarter, an eighth, etc) and comparing the distributions of avalanche sizes and durations by computing the average pairwise KS-distance between all four distributions. We then rebinned the binary time series with the inverse of the fraction of the sample selected (eg: when subsampling half the nodes, we rebinned the timeseries by two, when sampling a quarter, rebinned by four, etc).

When compared, all conditions showed visual indicators of finite size scaling collapse (see Fig 9 for an illustration with the distributions of avalanche sizes). To quantify this, we calculated the fold-change between the average pairwise KS-distance before and after re-binning. For the distributions of avalanche sizes, ketamine showed the highest average fold change

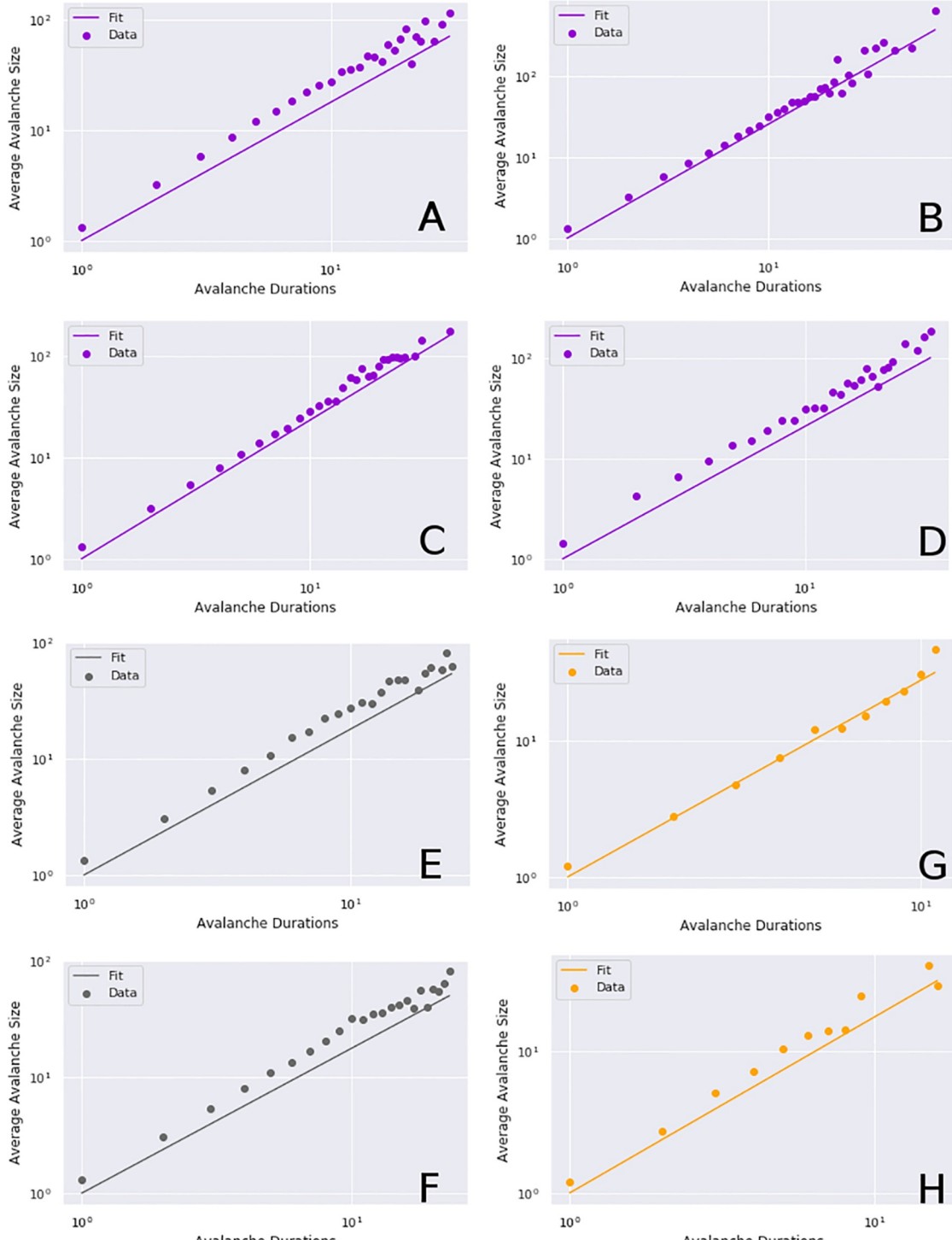

**Fig 8. The average avalanche size for a given duration by condition.** The color mapping is the same as in Fig 7: purple corresponds to the awake condition (**A, B, C, D**), grey to ketamine (**E, F**), and orange to propofol (**G, H**). Each plot includes two scans: the anaesthesia condition (propofol or ketamine) and the associated pre-anesthesia awake condition. In all cases there is a clear linear relationship between the average avalanche size for a given duration. Such a relationship is considered necessary, but not sufficient, to conclude a system is displaying critical dynamics [17].

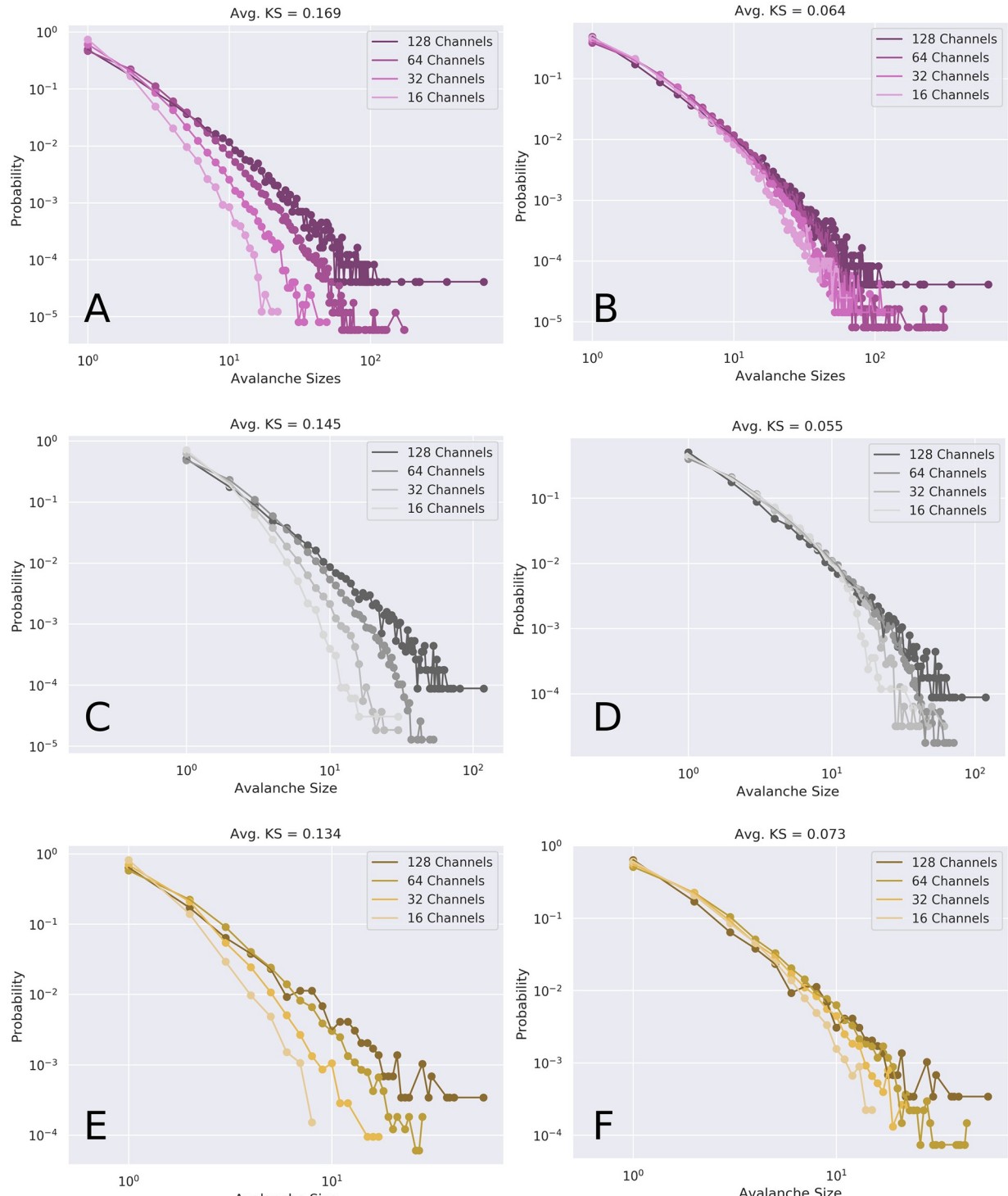

**Fig 9. Data collapse under rescaling.** Shape collapse for the various subsampled avalanche sizes before (left column) and after (right column) renormalizing (rebinning) the binary timeseries. The color indicators are consistent with other figures: purple is awake (**A, B**), grey is ketamine (**C, D**), yellow is propofol (**E, F**). In all conditions, renormalization resulted in noticeable shape collapse, which can be quantified by doing pairwise calculations of the Wasserstein distance metric. Unlike Figs 7 and 8, this does not show all scans, but rather, representative examples of shape collapse for each of the three conditions.

(−0.66), indicating the greatest collapse. The awake condition came second with a fold-change of −0.61, and the propofol condition showed the weakest shape-collapse, with a fold change of −0.49. For the avalanche duration distributions, the pattern was similar, with ketamine showing the greatest fold change (−8.1), but in this case, propofol was in the middle with a fold change of −0.73, and the awake condition (−0.69) at the end.

This suggests that all the conditions display some measurable collapse when renormalized, and in both avalanche shape and durations, the effect was most dramatic in the ketamine condition, compared to awake and propofol.

## Complexity

There were dramatic differences between the degree to which ketamine and propofol reduced the complexity of brain activity (Fig 10). Both anaesthetics reduced the total $C_N(X)$, however, propofol had a much stronger effect. On average, the fold-change between the awake condition and the ketamine condition was −0.35, compared to the awake vs. propofol condition, which showed an average fold-change of −0.91. This is a strong indicator that, even at surgical doses, ketamine supports significantly more complex brain dynamics than propofol.

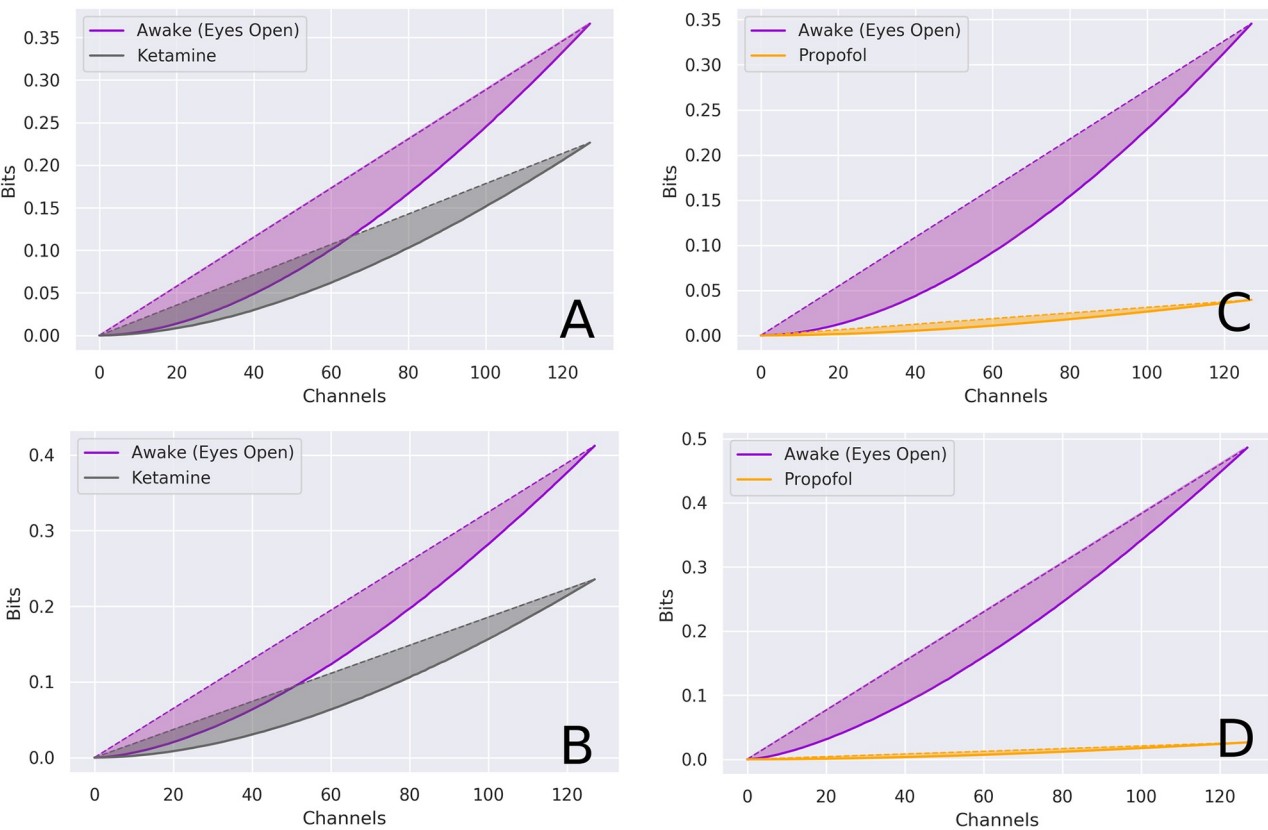

**Fig 10. Differences in multi-scale complexity between conditions.** Integration of information across scales for awake (purple) vs. ketamine (grey) (**A, B**) and awake vs. propofol (orange) (**C, D**). Complexity was calculated using the NCC Toolbox [48], using the measure first proposed by Tononi, Sporns & Edelman [41]. While the ketamine condition (grey) was associated with a noticable decrease in complexity compared the awake condition (purple), it was a much smaller decrease than what was observed in the propofol condition, which was an order of magnitude less complex. Note that the total value of complexity is not the greatest value, but rather, the difference between the area under the curves for the linear fit and the true non-linear integration.

## Discussion

In this manuscript we present several converging lines of evidence that propofol and ketamine produce different effects on markers of critical brain dynamics. Furthermore, ketamine, but not propofol, appears to supports dynamics similar to those observed in normal waking consciousness in ECoG activity recorded from a single macaque in different states of consciousness. Despite inducing states that appear similar to external observers (unresponsiveness to pain or sensory stimuli, decreased voluntary motor output, anaesthesia), propofol produces a near-total extinction of consciousness, while ketamine frequently induces states of "dissociative" anaesthesia, which can include dream-like or out-of-body experiences [36, 37]. To test for signs of criticality we explored the dynamics of avalanches (coordinated bursts of high-intensity activity) across the cortex, testing for scale-free distributions and exponent relations between them. We also examined universality under renormalization and an information-theoretic measure of multi-dimensional complexity to further characterize the effects of these drugs on brain dynamics.

We found that ketamine slightly reduced the rate of events across the cortex when compared the awake condition, while the propofol condition reduced activity rates by an entire order of magnitude. Within conditions, the distributions of activities across channels were consistent between scans, suggesting that these measures are stable across multiple drug experiences, at least within one individual.

### Avalanche distributions

We found that, when compared to normal wakefulness, propofol dramatically attenuated signs of power-law distributions in both the distributions of avalanche sizes and durations. Heavy tailed distributions imply that avalanche dynamics are playing out over a range of temporal and spatial scales; consequently, the finding of multiscale dynamics in the awake and ketamine conditions, but not the propofol condition suggests that this kind of multiscale coordination is significant for the maintenance of consciousness. This is consistent with previous work suggesting that loss of multiscale structure and decreases in the fractal dimension of EEG signals under sedation [58–62]. The spectral exponent of the power spectrum density function, which typically follows a power-law decay has been found to be strongly indicative of conscious states [62, 63], providing evidence that multi-scale, or scale-free processes in the brain are related to the maintenance of normal awareness.

The differences in the distributions of avalanche sizes and durations may be indicative of changes to the ability of the brain to coordinate activity in different states. In the awake condition, we observed avalanches that included many distinct channels, as well as being comparatively longer lived, while those long, large avalanches were significantly impaired by propofol, but not by ketamine. The propofol result is consistent with previous work which found that propofol inhibited the ability of the brain to form integrated higher-level networks, while leaving local activity in tact [64, 65]. If avalanches indicate when signals are able to propagate across the cortex, we might expect to see alterations in avalanche activity being associated with fragmentation or disintegration of functional connectivity networks. Previous research using point-processing methods on fMRI time-series has found that rare, high-amplitude events can capture a significant information relevant to critical brain dynamics and states of consciousness [47, 66]. Using techniques from information theory such as mutual information and transfer entropy [67] it would be informative to construct functional connectivity networks from the binarized time-series to directly relate changes in critical dynamics to alterations to network topologies and computational capabilities [68, 69].

The relative differences in the frequency of large avalanches between the three conditions may also be relevant to work with the Global Workspace Theory (GWT). Previous work has proposed that large avalanches represent the "ignition" of information percolating through the neuronal global workspace [70, 71]. In the context of the GWT, this percolation is thought to define the difference between information processing that is "conscious", from processing that is "unconscious" [72], and so ketamine may allow consciousness to persist (in contrast to propofol) by failing to disrupt the ability of information to "ignite" and propagate into the Global Workspace.

## Indicators of critical dynamics

The other indicators of criticality are harder to interpret. All conditions showed similar patterns of critical exponent relations, with the value of $1/\sigma v z$ calculated from the avalanche shape collapse showing a high degree of agreement when calculated from the regression of the average avalanche size for a given avalanche duration. In contrast, both values of $1/\sigma v z$ showed less consistency when compared to the exponent relation $\alpha - 1/\tau - 1$. Finally, when individual channels were subsampled and rescaled, all conditions showed reasonable signs of collapse, with ketamine showing the most significant signs of operating near the critical point.

None of the states showed consistently tighter exponent relations, although we should stress that all exponents were calculated *within the range of $x_{min}$ and $x_{max}$ for which the power-law hypothesis fit with p-values $\geq 0.2$*. The propofol condition had a dramatically reduced range of $x_{min}$ and $x_{max}$ values compared to awake and ketamine, so whatever inferences about the plausibility of critical dynamics we make from these data is only relevant within these ranges. Consequently, although all three conditions showed reasonably similar behaviour in terms of exponent relations, overall, the awake and ketamine conditions showed this behaviour over 2 orders of magnitude, while propofol showed it over one tenth of that range. Based on these, we propose that the brain is able to support critical dynamics in all three states, but that propofol (but not ketamine) *reduces the scale over which critical dynamics can occur*. Critical dynamics are often described as being 'scale-free', so the notion of restricting critical dynamics to a range of scales may seen contradictory. However perfect scale-freeness of critical systems only occurs in infinite systems. In all finite systems, criticality can emerge in a restricted range. It is worth unpacking how this restriction might play out in the brain. Two dynamical changes might restrict the range over which the power-law held. An increase in high-frequency noise will have the effect of driving a deviation from power-law scaling at the upper end of the distribution, while limiting the diverging correlation length will drive a deviation from power-laws at the lower end of the tail. By examining how the $x_{min}$ and $x_{max}$ values change between conditions, we can better understand the changing dynamics. In the propofol condition, for both avalanche sizes and durations, the values of $x_{min}$ and $x_{max}$ are shifted up the distribution, so the power-law fit begins and ends with smaller, shorter lived avalanches. This could be consistent with both a reduction in high-frequency noise, as well as a decreasing correlation length, both of which are consistent with other, well-established elements of propofol anaesthesia. As previously mentioned, a leading hypothesis is that anaesthetics like propofol "fragment" brain networks [64, 73, 74], or alternately "mute" the flow of information [75] between regions. In the context of functional connectivity analysis (a core element of many of these analyses), decreased connectivity can be directly related to a decrease in correlation length, which is consistent with the loss of power-law behaviour in the tails of the distributions. The decrease in high-frequency noise is likely associated with the increase in high-power, low-frequency oscillations that characterize the state induced by propofol [76, 77].

This shrinking of the critical zone may provide an explanation for the original motivating question for this study: why do critical dynamics seem preserved in unconscious systems at the micro-scale [16, 17, 27], but are altered at the macroscale [26, 62]? It may be that at the local level, neural networks are able to maintain critical dynamics, but the ability of these local ensembles to coordinate is impaired [64, 65]. Relevant to this hypothesis is the finding that anaesthetics alter the functioning of pyramidal neurons in Layer IV and V of the cortex [78, 79]. As Layer V pyramidal neurons are thought to serve as "outputs" from one brain region to another [80] and so changes to pyramidal neural functioning may explain how local critical dynamics are able to propagate to macro-scale critical dynamics. Layer V pyramidal neurons expressing 5-HT$_{2A}$ receptors have already been implicated in macro-scale critical dynamics [25] in the context of serotonergic drugs, so a possible involvement in anaesthesia is not totally out of the question.

The question about relative ranges is significant for the exponent relationship results, but not necessarily for the universality and sub-sampling results, which are probably the hardest to interpret. All three conditions showed roughly equivalent degrees of shape collapse upon time-series re-binning, suggesting that all three display self-similar dynamics across channels. One possible interpretation is that, despite the alteration in consciousness induced by propofol and ketamine, the brain is able to adaptively maintain at least some qualities of critical dynamics. Previous work has found that the brain appears to adapt to perturbations and return to the critical regime despite alterations to incoming sensory inputs [16, 81]. One possible explanation for the persistence of signs of criticality is that the brains are rapidly adapting to the drug state. If this is the case, it presents an intriguing window of possibility: might it be possible to break "criticality" writ-large down into separate phenomena and explore which ones are necessary (or sufficient) for complex consciousness and cognition, and which may be irrelevant?

## Complexity

We also found that, using a multi-scale measure of complexity [41], the propofol condition was associated with dramatic decreases in the complexity of brain activity, while ketamine preserved higher levels of complexity.

The most dramatic difference is the effect of the anaesthetics on multi-scale complexity. This measure has been suggested to be relevant to the emergence of phenomenological consciousness [82]. Our findings are consistent with tendency of ketamine to preserve consciousness in states of dreamlike "dissociative anaesthesia" in contrast to propofol. It is unsurprising that highly complex behavioural states should be underpinned by complex dynamics, and evidence from studies of anaesthesia [83–88], disorders of consciousness [89–91], sleep [92], and psychedelic states [25, 26] bears this out. Our results are consistent with, and extend these previous findings using a more principled measure of "complexity" than randomness-based measures.

## Limitations

One of the most significant limitations of this work is that it rests on the assumption that, when in the ketamine condition, the monkey was experiencing dissociative anaesthesia, as opposed to true anaesthesia (as it presumably experienced under propofol). The dosage used in the original study (5.1mg/kg) were consistent with doses used for pre-surgical anaesthesia in primates [32] and it is unclear at what dose dissociative anaesthesia transitions into "total" anaesthesia (with no dream-like content at all). Presumably even in cases of "true" ketamine anaesthesia, the subject passes through hallucinatory states during induction and emergence [35–37]. It is also not entirely clear that, even if the dose was appropriate to induce

hallucinatory states of dissociative anaesthesia in humans, whether macaques experience such a state at all. Despite this limitation, the fact that ketamine maintained wakefulness-like dynamics on multiple measures implies that it is not inconceivable that whatever processes allow consciousness to persist under ketamine in humans are also playing out in the macaque brain.

There is also a question about how the difference between the open eyes in the awake condition, compared to the closed eyes in the anaesthesia conditions, may be affecting neural dynamics. It could be argued that the awake state may be less stationary than either anaesthesia states, as the monkey is still able to engage with it's environment despite the restraints. The documentation does not make it clear how much potentially stimulating activity was taking place during the period of awake recording, although the monkey is described as "calmly sitting for periods up to 20 minutes" (http://wiki.neurotycho.org/Anesthesia_and_Sleep_Task_Details). While this may be considered a limitation, we argue that alertness and responsiveness to the environment is actually a key component to understanding "normal" waking consciousness. Awareness of, and responsiveness to, the environment is a key element of what it means to be conscious, and so while the effects of incoming stimuli on neural data remain a fascinating outstanding question, we do not think that their presence is incompatible with our goal of understanding the differences between these three states of consciousness.

Having only a single viable monkey to analyse is another significant limitation, although given the rarity of this dataset, we believe the results are still informative. This work should, however, be replicated in a larger cohort of humans or animals undergoing similar anaesthesia and recording procedures. Concerns about the low power are at least somewhat ameliorated by the fact that, for our monkey, we have 2× scans in both the propofol and ketamine conditions, and 4× scans in the awake condition. In general, the results were quite consistent within conditions, suggesting that, at least for this individual, the effects observed here are reasonably robust.

Finally, the analyses described here require binarizing continuous time-series data, which throws out a considerable amount of information (although point-processes like this have been found retain surprising amounts of information, see [47, 66]). For this study, the point process was necessary as the indicators of critical dynamics we explored here were derived from models of binary branching processes [17]. Future analyses that incorporate the full, continuous time series may provide insights missing from this current body of work. The method that we used to define an "event" (an above-threshold excursion in one channel or a sequence in another channel that is significantly correlated with the excursion) partially addresses this concern by including temporal similarity as a criteria by which an event could be identified, but further improvements are no doubt possible.

## Conclusions

These results show that, despite their shared status as anaesthetics and the similarity of the external appearance of effects, propofol and ketamine cause different brain states typified by dramatically different neural dynamics. Unlike propofol, ketamine allows for large, long-lasting avalanches of coordinated activity to persist in the cortex in a manner similar to normal consciousness, as well as maintaining a much higher degree of multi-scale complexity. These results may explain why the state induced by ketamine can often include complex, phenomenological consciousness to persist in a dissociated, dream-like state, while propofol extinguishes awareness completely. Understanding which brain dynamics are necessary or sufficient to support consciousness is of interest to both for those interested in the theoretical

basis of conscious awareness as well as addressing clinical concerns related to identifying covert consciousness in patients who may not be able to directly communicate.

## Glossary

**Anaesthetic**: A drug that reversibly disrupts normal consciousness, inducing a state of coma-like unconsciousness.

**Criticality**: Refers to a system on the boundary between two qualitatively distinct phases. A canonical example from physics is boiling water, which is at the critical boundary between liquid and gas.

**Metastability**: Refers to a dynamical regime where the system orbits a set of competing attractors without any one "winning." Associated with the maintenance of complex, adaptive behaviour.

**Power Law**: A statistical distribution where $P(X = x) \propto x^{-\alpha}$. Critical systems are expected to produce power law distributions of events, although non-critical systems can produce them as well.

**Universality**: When a complex system has the same critical exponents even when smaller scale details are varied, it is said to belong to a universality class.

## Supporting information

**S1 Table. All of the results referred in Table 1 as a.csv file.**
(CSV)

**S1 Scripts. Scripts required to recreate data analysis and figures.**
(TAR.GZ)

## Acknowledgments

We would like to thank Dr. Yang-Yeol Ahn for advice on power-law inference and visualization, Dr. Filippo Radicci for insights and discussion, as well as Joshua Faskowitz and Dr. Alice Patania for support, friendship, and advice.

## Author Contributions

**Conceptualization:** Thomas F. Varley, Olaf Sporns, John Beggs.

**Data curation:** Thomas F. Varley, Aina Puce.

**Formal analysis:** Thomas F. Varley.

**Methodology:** Thomas F. Varley, Aina Puce, John Beggs.

**Software:** Thomas F. Varley.

**Supervision:** Olaf Sporns, John Beggs.

**Visualization:** Thomas F. Varley, Olaf Sporns.

**Writing – original draft:** Thomas F. Varley.

**Writing – review & editing:** Thomas F. Varley, Olaf Sporns, Aina Puce, John Beggs.

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
