## [Decision Letter · Decision Letter 0]

24 Jun 2020

Hi

The three reviewers raise a number of concerns, but two of them are particularly salient:

(1) is the work sufficiently novel.

(2) can conclusions be drawn from such a small sample size?

I would like to invite the authors to convincingly address these concerns alongside the other points raised by the reviewers.

Best wishes

Saad

---------------

Dear Mr Varley,

Thank you very much for submitting your manuscript "Differential Effects of Propofol and Ketamine on Critical Brain Dynamics" for consideration at PLOS Computational Biology.

As with all papers reviewed by the journal, your manuscript was reviewed by members of the editorial board and by several independent reviewers. In light of the reviews (below this email), we would like to invite the resubmission of a significantly-revised version that takes into account the reviewers' comments.

We cannot make any decision about publication until we have seen the revised manuscript and your response to the reviewers' comments. Your revised manuscript is also likely to be sent to reviewers for further evaluation.

Sincerely,

Saad Jbabdi

Associate Editor

PLOS Computational Biology

Kim Blackwell

Deputy Editor

PLOS Computational Biology

Reviewer's Responses to Questions

**Comments to the Authors:**

Reviewer #1: Using ECoG recordings in a macaque brain, this study investigated the effect of different anesthetics (propofol and ketamine) on critical brain dynamics. Consistent with different mechanisms of the two anesthetics, they found propofol, but not ketamine, dramatically restricted the size and duration of avalanches, as well as a large reduction in the complexity of brain dynamics. Overall, I think this is a well-conducted study. The methodology is sound, the conclusion helps to improve our understanding of altered states of consciousness and brain dynamics.

My main concern is about the small sample size. I agree with the authors about the rarity of the dataset, but with only N=4 for wake sessions and N=2 for propofol and ketamine, I think the results with mean+/-SD is inappropriate and somewhat misleading. I would suggest changing the descriptive statistics in the main text (perhaps only mean, or the range like minimum-maximum?), and moving the individual session results in supplemental table 1 to main text for clarity.

Line 122. Please clearly specify the number of scans for each condition (wake, propofol, ketamine).

Line 132. Please specify the sampling frequency of the data.

Line 156-157. The description of the method is not clear. For example, which correlation function was used? Is ρ correlation coefficient? What do t_min and t_max mean?

Line 192. So, x denotes avalanche size or duration, and minimum and maximum values of x, which were named as x_min and x_max in line 234? Please clarify.

Line 319. For avalanche size distribution, with only 128 channels of ECoG, how to get the maximum values of 378? Was this derived from fitted data or empirical data?

Figures 7, 8 and 10 showed the results from two scans? Please appropriately describe this in the legend.

Line 516. I would suggest a brief discussion on the effect of eyes open on awake results, and on the observed difference between awake and unresponsive conditions (propofol/ketamine).

Reviewer #2: The authors studied the differential effects of propofol and ketamine on the critical brain dynamics of a single macaque. A previously validated test of criticality, avalanche dynamics, was applied to analyze ECoG data of propofol and ketamine that induce differentiable effects on consciousness. Many previous studies with human and animal subjects suggested that maintenance of critical dynamics is necessary for the emergence of consciousness. However, controversially, some studies also demonstrated criticality in unconscious states. Thus, the authors tried to fill the knowledge gap and showed that propofol dramatically restricted the size and duration of avalanches, while ketamine allowed for more awake-like dynamics to persist. And propofol produces a dramatic reduction in the complexity but all states show some signs of persistent criticality. The author concluded that maintenance of critical brain dynamics may be important for regulation and control of conscious awareness.

The paper was well written and published timely. The authors provided proper background and relevant knowledge. The novelty and the need for this study appear clearly. I think this study may provide many insights to the researchers in this research field especially on the controversial findings of criticality in conscious and unconscious states.

Despite this paper was well written, I feel this study has a serious limitation that has already been mentioned by the authors. That is, all the results were drawn only from a single monkey. Even though several ECoG recordings were analyzed for two different anesthetics, still I doubt whether the results could be reproducible with other subjects. Considering the ambiguity of determining consciousness which mainly depends on the subject’s responsiveness, in particular, with monkey it is worse than human subject. The authors cannot completely get rid of the possibility that the macaque was in covert consciousness. The subject could have covert consciousness if the dose of propofol was not enough to induce deep anesthesia or the anesthetic concentration was not maintained during the ECoG recording (in this study, the concentration was not maintained), or if the level of consciousness fluctuated after the inintal induction, the subject could have woken up for a moment. Because of these possibilities, the authors did not remove, it seems difficult the authors can conclude that the signs of criticality were observed in the unconscious state. Without gatering more subjects, it would be difficult to improve this critical limitation.

Reviewer #3: Varley and colleagues investigated the properties of non-human primate ECoG data from the perspective of critical dynamics, with the hypothesis that consciousness (perhaps more adequately, responsiveness) would correlate with critical behaviour. They found some evidence supporting that propofol (but not ketamine) disrupts critical dynamics.

While this direction of research is interesting, the results are hardly novel at this stage. The authors cited several papers showing a departure from critical dynamics induced by general anesthetics, including propofol. This extends to other states of reduced consciousness, such as slow wave sleep or epileptic seizures. Perhaps the most interesting finding is that ketamine resembled wakefulness; although this could also be expected from previous work, I'm not aware it has been shown from the perspective of criticality.

Besides this potential novelty issue, I have the following comments for the authors:

- The use of the word "exotic" to refer to non-ordinary/altered states of consciousness is somewhat strange... is there a reason not to stick to the common nomenclature?

- I checked ref. 25 and it does not include analysis of ECoG recordings acquired under the effect of psychedelics - this would have been somewhat strange, I think, since invasive recordings tend to be acquired in neurological patients only.

- The NeuroTycho dataset has ECoG data acquired during other potentially interesting states of reduced awareness, which, if analyzed by the authors, could contribute to broaden the scope of their conclusions. I'm thinking of the sleep and ketamine plus medetomidine datasets. Is there a reason to exclude these datasets and to focus on the ketamine and propofol recordings only?

- The following hypothesis "propofol would dramatically reduce signs of critical dynamics, but that criticality would persist under the influence of ketamine" is reasonable for sub-anesthetic doses of ketamine, but that is not so clear for anesthetic doses (at least not from the previous paragraphs of the introduction)

- Perhaps the authors should invest additional efforts in model comparison, for instance, how do log normal, exponential, and exponential cutoff models reproduce the avalanche distributions, and how do the goodness of fit compare to those seen for power laws?

- "To avoid interminable run-times, the NCC Toolbox [49] includes several corrections for sub-sampling and heuristics for estimating integration in large systems" -> Maybe some information concerning those heuristics, to make the manuscript more self-contained?

- "Visual inspection shows that theketamine condition tracks the Awake condition much more closely than the propofol condition does" -> Sorry, I really can't see this by visual inspection. I know the authors checked the stats on the manuscript text, but perhaps they should make one or more new figures where these are visualized. For example, the values of the exponents are scattered throughout the text and this makes it difficult to draw quick comparisons. Here a figure could help.

- "We had hypothesized that the Awake and/or ketamine conditions would show the highest degree of concurrence between the measures (reflecting a greater degree of criticality), but instead, the Awake condition has the lowest degree of concurrence" -> I think this deserves more discussion.

- "Based on these, we propose that the brain is able to support critical dynamics in all three states, but that propofol (but not ketamine) reduces the scale over which critical dynamics can occur." -> At this point, I think you need to delve deeper into how this "partial" scale-free dynamics relate to known results from statistical physics. What kind of system would show critical dynamics over a restricted range, and is the brain such a system?

**Have all data underlying the figures and results presented in the manuscript been provided?**

Reviewer #1: Yes

Reviewer #2: None

Reviewer #3: Yes

PLOS authors have the option to publish the peer review history of their article (what does this mean?). If published, this will include your full peer review and any attached files.

Reviewer #1: **Yes: **Duan Li

Reviewer #2: No

Reviewer #3: No
---

## [Decision Letter · Decision Letter 1]

5 Oct 2020

Dear Mr Varley,

We are pleased to inform you that your manuscript 'Differential Effects of Propofol and Ketamine on Critical Brain Dynamics' has been provisionally accepted for publication in PLOS Computational Biology.

Best regards,

Saad Jbabdi

Associate Editor

PLOS Computational Biology

Kim Blackwell

Deputy Editor

PLOS Computational Biology

Reviewer's Responses to Questions

**Comments to the Authors:**

Reviewer #1: The authors have addressed my comments appropriately.

Reviewer #2: I think the authors well explained and applied my concerns to the revised manuscript.

Reviewer #3: The authors have addressed all my concerns extensively and adequately, and because of this I recommend this manuscript for publication.

**Have all data underlying the figures and results presented in the manuscript been provided?**

Reviewer #1: None

Reviewer #2: None

Reviewer #3: Yes

PLOS authors have the option to publish the peer review history of their article (what does this mean?). If published, this will include your full peer review and any attached files.

Reviewer #1: **Yes: **Duan Li

Reviewer #2: No

Reviewer #3: **Yes: **Enzo Tagliazucchi

---

## [Editor Report · Acceptance letter]

24 Nov 2020

PCOMPBIOL-D-20-00547R1 

Differential Effects of Propofol and Ketamine on Critical Brain Dynamics

Dear Dr Varley,

I am pleased to inform you that your manuscript has been formally accepted for publication in PLOS Computational Biology. Your manuscript is now with our production department and you will be notified of the publication date in due course.

With kind regards,

Nicola Davies
